# Synergism of primary and secondary interactions in a crystalline hydrogen peroxide complex with tin

Alexander G. Medvedev [1], Pavel A. Egorov [1], Alexey A. Mikhaylov [1], Evgeny S. Belyaev [2], Gayane A. Kirakosyan [1,2], Yulia G. Gorbunova [1,2], Oleg A. Filippov [3], Natalia V. Belkova [3], Elena S. Shubina [3], Maria N. Brekhovskikh [1], Anna A. Kirsanova [4], Maria V. Babak [4] ✉, Ovadia Lev [5] ✉ & Petr V. Prikhodchenko [1] ✉

Despite the significance of $H_2O_2$-metal adducts in catalysis, materials science and biotechnology, the nature of the interactions between $H_2O_2$ and metal cations remains elusive and debatable. This is primarily due to the extremely weak coordinating ability of $H_2O_2$, which poses challenges in characterizing and understanding the specific nature of these interactions. Herein, we present an approach to obtain $H_2O_2$–metal complexes that employs neat $H_2O_2$ as both solvent and ligand. $SnCl_4$ effectively binds $H_2O_2$, forming a $SnCl_4(H_2O_2)_2$ complex, as confirmed by $^{119}Sn$ and $^{17}O$ NMR spectroscopy. Crystalline adducts, $SnCl_4(H_2O_2)_2 \cdot H_2O_2 \cdot 18$-crown-6 and $2[SnCl_4(H_2O_2)(H_2O)] \cdot 18$-crown-6, are isolated and characterized by X-ray diffraction, providing the complete characterization of the hydrogen bonding of $H_2O_2$ ligands including geometric parameters and energy values. DFT analysis reveals the synergy between a coordinative bond of $H_2O_2$ with metal cation and its hydrogen bonding with a second coordination sphere. This synergism of primary and secondary interactions might be a key to understanding $H_2O_2$ reactivity in biological systems.

Hydrogen peroxide, a highly stable reactive oxygen species[1], is widely recognized for its vital contributions to diverse cellular functions including protection against oxidative stress, promotion of cellular differentiation, facilitation of cellular proliferation, and participation in redox signaling pathways[2–4]. Furthermore, owing to its oxidative properties, $H_2O_2$ holds significant importance in a range of industrial applications, including bleaching processes, wastewater treatment and various catalytic processes utilized in industry. While $H_2O_2$ can be employed in a metal-free processes the effectiveness and selectivity of $H_2O_2$ as an oxidant can be further enhanced through its kinetic activation by metal complexes[5,6]. For example, the activation of $H_2O_2$ by its

coordination to heme enables the efficient utilization of $H_2O_2$ in various biochemical reactions catalyzed by cytochrome P450 heme-containing enzymes[7]. Despite the fundamental importance of these interactions, their molecular mechanism remains unclear due to the transient and labile nature of $H_2O_2$-metal adducts.

If we consider complexation as a reaction between Lewis acids and bases, the coordination ability of a ligand may be correlated with its basicity, which can be characterized in terms of its basicity constant ($pK_b$, which is often substituted by the $pK_a$ of the ligand's corresponding conjugate acid) or proton affinity (PA; that is, the enthalpy of the $B + H^+ \rightarrow BH^+$ reaction, where B = a base). As the PA of $H_2O_2$ is

[1]Kurnakov Institute of General and Inorganic Chemistry, Russian Academy of Sciences, Moscow, Russian Federation. [2]Frumkin Institute of Physical Chemistry and Electrochemistry of the Russian Academy of Sciences, Moscow, Russian Federation. [3]Nesmeyanov Institute of Organoelement Compounds, Russian Academy of Sciences, Moscow, Russian Federation. [4]Drug Discovery Lab, Department of Chemistry, City University of Hong Kong, Kowloon, Hong Kong SAR, China. [5]Casali Center of Applied Chemistry, Hebrew University of Jerusalem, Jerusalem, Israel. ✉e-mail: mbabak@cityu.edu.hk; ovadia@mail.huji.ac.il; prikhman@gmail.com

4 kcal mol$^{-1}$ less than that of $H_2O$[8], the coordination of $H_2O_2$ at low concentrations with a metal center is thermodynamically unfavorable in aqueous solutions. Indeed, Williams et al. reported that $H_2O_2$ complexes are thermodynamically unstable in aqueous solution unless the pH is sufficiently high to deprotonate $H_2O_2$ and thus favor the formation of hydroperoxo coordination compounds[9]. Subsequently, DiPasquale and Mayer demonstrated that $H_2O_2$ does not displace a very weakly bound perchlorate ligand from the gallium(III) center of a tetraphenylporphyrin complex[10]. Thus, as formulated by Mayer, $H_2O_2$ typically exhibits poor coordination ability due to its rather low PA, which is attributable to its electron-withdrawing hydroxyl (-OH) group being adjacent to an O atom[10]. A few examples of $H_2O_2$ coordinated with cobalt(II), nickel(II), and copper(II) in non-aqueous solutions were recently reported, supported by nuclear magnetic resonance (NMR) and cyclic voltammetry data[11,12]. In addition, many aqua complexes have been identified in solution and solid forms, but only one complex of $H_2O_2$ with a metal cation has been isolated and structurally characterized, namely a complex of $H_2O_2$ and zinc(II)[13]. However, the isomorphic substitution of $H_2O_2$ with $H_2O$ (in a 50:50 occupancy ratio) disordered the O atoms and tosyl fragments in this complex, preventing the positions of the $H_2O_2$ protons being determined objectively. In contrast to transition metals, p-block elements do not catalyze $H_2O_2$ decomposition. For example, tin compounds are known as $H_2O_2$ stabilizers as $Sn^{IV}$ forms stable hydroperoxo complexes with high peroxide content[14]. However, to the best of our knowledge, the coordination of $H_2O_2$ with $Sn^{IV}$ has not been reported.

Octahedral coordination is typical for $Sn^{IV}$, but coordinatively unsaturated tin tetrachloride ($SnCl_4$) is a strong Lewis acid and thus we hypothesized that it can bind $H_2O_2$ in the absence of ligands of higher basicity. We confirmed this hypothesis by performing $^{119}Sn$ and $^{17}O$ NMR studies that characterized $H_2O_2$ coordination by $SnCl_4$ and $H_2O_2$ substitution by $H_2O$, methanol (MeOH), and acetonitrile (MeCN) in the $Sn^{IV}$ coordination sphere. In addition, we performed single-crystal X-ray diffraction (scXRD) analysis of two crystalline adducts of $H_2O_2$ with 18-crown-6, namely $[SnCl_4(H_2O_2)_2]\cdot H_2O\cdot 18\text{-crown-}6$ (1) and $2[SnCl_4(H_2O_2)(H_2O)]\cdot 18\text{-crown-}6$ (2), which enabled examination of the intermolecular interactions in these structures. Moreover, we performed density functional theory (DFT) modeling to unveil the synergy between various types of bonds in which $H_2O_2$ is engaged in 1 and 2 and how this effect stabilizes these complexes.

## Results and discussion
As $H_2O_2$ is less basic than other polar solvents[8] and it does not form homogeneous solutions with non-coordinating solvents, we used neat $H_2O_2$ both as a ligand and as a solvent to study its interaction with $SnCl_4$.

### $^{119}Sn$ and $^{17}O$ NMR studies
The transformation of the $Sn^{IV}$ coordination sphere upon addition of $H_2O_2$ proposed in Fig. 1A was studied by $^{119}Sn$ and $^{17}O$ NMR spectroscopy (Fig. 1B, C). The coordinatively unsaturated environment of $Sn^{IV}$ in neat $SnCl_4$ was revealed by its low-field signal in the $^{119}Sn$ NMR spectrum ($\delta = -150$ ppm; Fig. 1Ba). The $^{17}O$ NMR spectrum of anhydrous $H_2O_2$ had a single signal at 180 ppm and no signals in the region of $H_2O$, i.e., at approximately 0 ppm, confirming that it contained less than 0.5 wt.% $H_2O$ (Fig. 1Ca). The addition of 1 wt.% $H_2O$ resulted in the appearance of a signal in the $^{17}O$ NMR spectrum at −5 ppm that was 0.8% of the integrated intensity of the $H_2O_2$ signal in this spectrum (Fig. 1Cb).

Careful addition of up to a fourfold molar excess of anhydrous $H_2O_2$ to $SnCl_4$ yielded a biphasic mixture (Supplementary Movie 1), whereas addition of a fivefold molar excess of anhydrous $H_2O_2$ (1:1 v/v) to $SnCl_4$ yielded a homogeneous mixture. $^{119}Sn$ NMR spectroscopy of this $SnCl_4–5H_2O_2$ system revealed a new high-field signal ($\delta_{Sn} = -554.7$ ppm) assigned to a $SnCl_4(H_2O_2)_2$ complex 1 (Fig. 1A, 1Bb) and a low-

intensity (3%) signal representing residual $SnCl_4$. The latter signal was absent in the $^{119}Sn$ NMR spectra of mixtures containing higher $H_2O_2$ concentrations (Fig. 1Bc). An $H_2O_2$ ligand in complex with tin contains two non-equivalent neighboring O atoms and thus its $^{17}O$ NMR signals are quadrupole broadened and therefore not detectable. As such, only one signal was present in the $^{17}O$ NMR spectrum ($\delta_O = 185.3$ ppm) and was assigned to free $H_2O_2$ (Fig. 1Cc). In the $^{119}Sn$ NMR spectra, a previously small signal at approximately −562 ppm became larger as the $H_2O_2$-to-$SnCl_4$ ratio increased (to 9:1; Fig. 1Bc) and was assigned to the complex bearing a bridging aqua ligand $[SnCl_4(H_2O_2)]_2(\mu\text{-}H_2O)$ (1′; Fig. 1A). Addition of $H_2O$ to give a 0.5 $H_2O$-to-$SnCl_4$ molar ratio resulted in $[SnCl_4(H_2O_2)]_2(\mu\text{-}H_2O)$ being the dominant species (Fig. 1Bd), and its $^{17}O$ NMR spectrum contained slightly upfield-shifted signals for free $H_2O_2$ ($\delta_O = 183.9$ ppm) and a bridging aqua ligand signal ($\delta_O = 51.3$ ppm; Fig. 1Cd). This conforms to $^{119}Sn$ NMR being very sensitive to changes in both the first and second coordination sphere of $Sn^{IV}$, so the subtle changes in the general composition of $Sn^{IV}$ complexes cause shifts of corresponding signals in $^{119}Sn$ NMR spectra.

Subsequent addition of another half equivalent of $H_2O$ resulted in the appearance of a high-field signal in $^{119}Sn$ NMR ($\delta_{Sn} = -589.7$ ppm; Fig. 1Be) and a new high-field $H_2O$ signal ($\delta_O = 31.9$ ppm) in the corresponding $^{17}O$ NMR spectrum (Fig. 1Ce) that we assigned to complex $SnCl_4(H_2O_2)(H_2O)$ (2; Fig. 1A). This substantial change in the chemical shift of the O atom of a coordinated $H_2O$ suggested that it had changed from a bridging coordination mode to a terminal coordination mode. A gradual increase in the $H_2O$ concentration (to a threefold molar excess of $H_2O$ relative to $SnCl_4$) led to the appearance of new high-field signals in the $^{119}Sn$ NMR spectra, indicating the complete substitution of $H_2O_2$ ligands in the coordination sphere of $Sn^{IV}$ in the original complex to form $[SnCl_4(H_2O)]_2(\mu\text{-}H_2O)$ (2′; $\delta_{Sn} = -610.7$ ppm; Fig. 1Bf) and then the known[15] $SnCl_4$ diaqua complex $SnCl_4(H_2O)_2$ (3; $\delta_{Sn} = -633.2$ ppm; Fig. 1Bg). The $^{17}O$ NMR spectrum of this compound exhibited the signals of an aqua ligand and $H_2O_2$ at $\delta_O = 27.4$ and 181.7 ppm, respectively (Fig. 1Cf). The single resonance for the aqua ligand indicates a rapid exchange between the solvent and the ligand.

$H_2O_2$ ligands can also be substituted by other donor molecules. For example, the addition of a threefold molar excess (based on Sn) of MeOH to a $SnCl_4–H_2O_2$ system resulted in the formation of $SnCl_4(MeOH)_2$, as confirmed by $^{119}Sn$ NMR ($\delta_{Sn} = -609.1$ ppm, Fig. 1Bh)[16]. Moreover, addition of MeCN to a solution of $SnCl_4$ in anhydrous $H_2O_2$ resulted in the immediate formation of a crystalline complex $SnCl_4(MeCN)_2$ (4, Supplementary Fig. 4), as confirmed by scXRD[17]. This ease with which $H_2O_2$ ligands can be substituted is consistent with the large difference between the PA of $H_2O_2$ (161.2 kcal mol$^{-1}$) and the PAs of $H_2O$, MeOH, and MeCN (165.2, 180.3, and 186.2 kcal mol$^{-1}$, respectively)[8].

### Synthesis and crystal structure of $SnCl_4–H_2O_2$ adducts with 18-crown-6
$H_2O_2$ always forms two hydrogen bonds, which stabilize crystalline adducts[18,19]. Similarly, an $H_2O_2$ ligand in a complex with zinc(II) was previously found to form hydrogen bonds with the proton-accepting tosyl groups of neighboring ligands[13]. Additionally, the formation of crystalline adducts of octahedral $SnCl_4(L)_2$ complexes bearing small ligands (e.g., diaqua and MeOH) and large organic molecules such as cyclodextrins, cucurbiturils, cryptands, and crown ethers was previously demonstrated[20]. In the current study, we examined whether $SnCl_4–H_2O_2$ systems can be stabilized by 18-crown-6 ether, because this compound is impervious to oxidation and contains six oxygen atoms, which can act as proton acceptors. Moreover, hydrogen bonding of $H_2O_2$ with 18-crown-6 ether was previously revealed by scXRD analysis of a corresponding peroxosolvate[21].

Accordingly, crystals of 1−3 were obtained from solutions of $SnCl_4$ in 99.9 wt.% $H_2O_2$ in the presence of 18-crown-6 with and without $H_2O$,

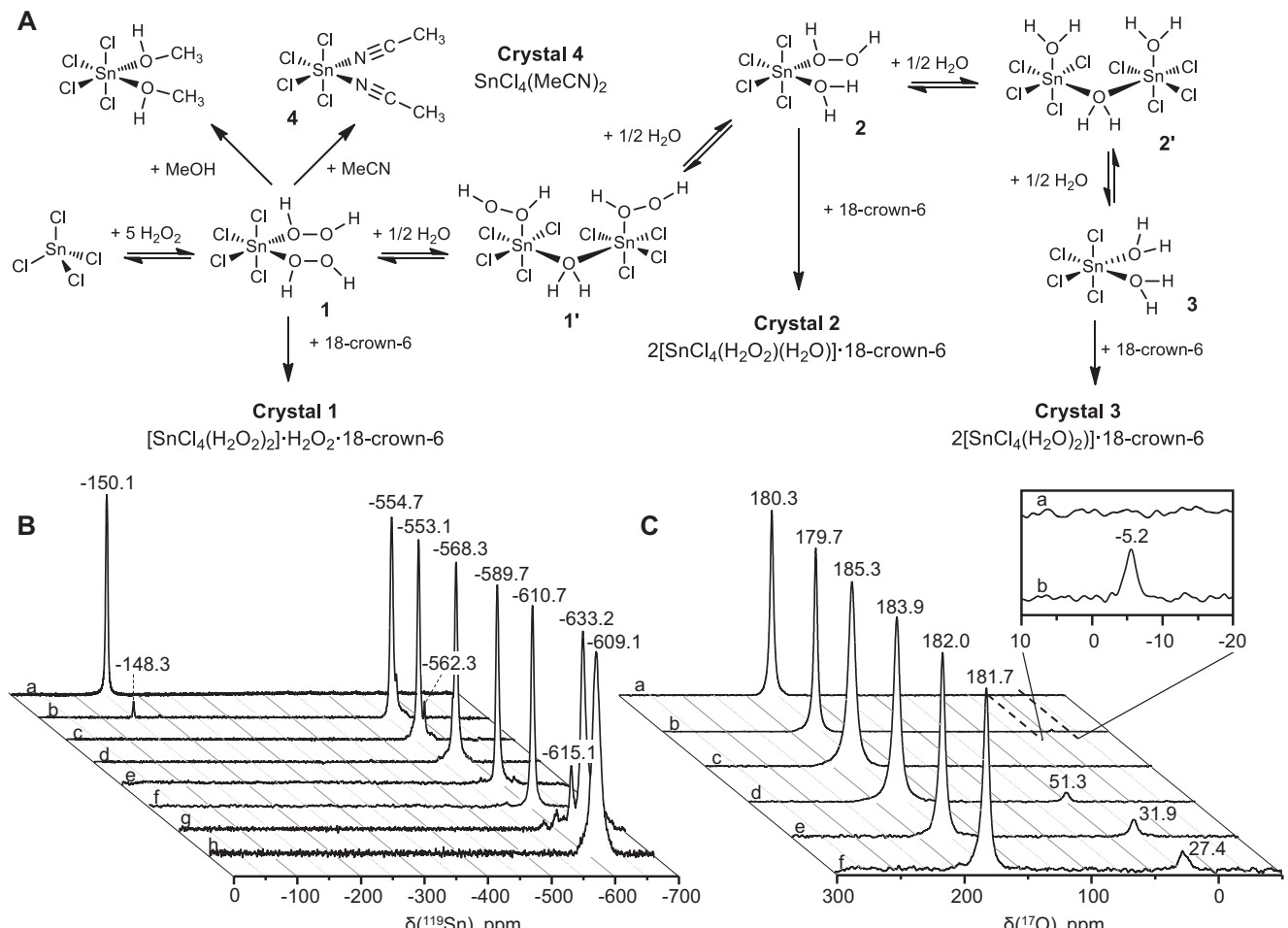

**Fig. 1 | Complexation of tin tetrachloride in hydrogen peroxide solution supported by NMR spectroscopy. A** Coordination of tin tetrachloride ($SnCl_4$) with hydrogen peroxide ($H_2O_2$), formation of crystalline compounds **1–4** and assumed intermediate complexes supported by [119]Sn and [17]O NMR. **B** [119]Sn nuclear magnetic resonance spectra of neat $SnCl_4$ (a); an $SnCl_4$–$H_2O_2$ (99.9%) system comprising a 1:5 molar ratio of $SnCl_4$ to $H_2O_2$ (b); 3M $SnCl_4$ in 99.9 wt.% $H_2O_2$ before (c) and after addition of 0.5 moles (d), 1 mole (e), 1.5 moles (f), and 3 moles of $H_2O$ with respect to Sn (g), and after addition of 3 moles of methanol with respect to Sn (h). **C** [17]O NMR spectra of 99.9 wt.% $H_2O_2$ before (a) and after addition of 1 wt.% of water ($H_2O$) (b); 3 M $SnCl_4$ in 99.9 wt.% $H_2O_2$ before (c) and after addition of 0.5 moles (d), 1 mole (e), and 3 moles of $H_2O$ with respect to Sn (f).

respectively, and analyzed by scXRD (Supplementary Table 1). This confirmed that **1–3** consisted of complexes with the compositions suggested by the NMR studies and unveiled a rich set of non-covalent interactions (Fig. 2 and Supplementary Fig. 1). In **1–3**, the $Sn^{IV}$ atom is present in a distorted octahedral environment with four chlorine atoms and two O atoms of $H_2O_2$ or $H_2O$ molecules, resulting in a *cis* isomer with O−Sn−O angles significantly less than 90° (Fig. 2A,B, Supplementary Table 2). The distances between the $Sn^{IV}$ and the O atoms of $H_2O_2$ (2.179(4) and 2.200(3) Å in **1**, and 2.225(3) Å in **2**) are much greater than those between the $Sn^{IV}$ and the O atoms of $H_2O$ (2.138(3) Å in **2**, and 2.133(2), 2.138(2) Å in **3**) (Table 1 and Supplementary Table 2). This reflects the weaker coordination of $H_2O_2$ to $Sn^{IV}$ than of $H_2O$ to $Sn^{IV}$, as confirmed by addition of $H_2O$ resulting in the substitution of $H_2O_2$ by $H_2O$. Moreover, the Sn−O distances in the complexes with aqua ligands exhibit a narrow range, but those in the complexes with $H_2O_2$ ligands exhibit a broader range. This disparity suggests that Sn−O interactions in the latter complexes are more significantly influenced by the strength of second-sphere hydrogen bonding than those in the aqua complexes. Thus, this hydrogen bonding fine-tunes the coordination of the $H_2O_2$ ligands.

Interestingly, the $H_2O_2$ ligands in the structures of **1** and **2** do not form hydrogen bonds as proton acceptors. This is similar to the $H_2O_2$ hydrogen bonding in a previously reported[13] $Zn^{II}$ complex and may be caused by coordination to the Lewis-acidic Sn species. Instead, the $H_2O_2$ and $H_2O$ ligands participate as proton donors in hydrogen bonding with crown ether molecules (**1–3**), a chlorine atom in the adjacent $SnCl_4$ fragment (**2**), and solvate $H_2O_2$ (**1**) (Supplementary Tables 3–5). The O(3)···O(5) distance in **1** (2.542(5) Å) and the O(1)···O(6) distance in **2** (2.548(4) Å) between the Sn-bound peroxo-OH moiety and adjacent $H_2O_2$ and ether oxygen, respectively, are much shorter than those in crystalline 18-crown-6 peroxosolvate (2.761(1)−3.040(1) Å)[21]. Furthermore, to the best of our knowledge, these distances are shorter than previously reported O···O distances in hydrogen bonds formed by $H_2O_2$ in crystalline peroxosolvates[20]. This suggests that the coordination of $H_2O_2$ with the Lewis acid ($Sn^{IV}$) in **1** and **2** results in an increase in the acidity of $H_2O_2$ that makes it form short hydrogen bonds. The phenomenon of hydrogen-bond enhancement due to coordination with Lewis acids has been observed in various hydrogen-bond donors[22,23] and widely applied in catalysis[24,25]. However, this phenomenon has not been reported to occur in complexes containing $H_2O_2$.

Binding to Sn leads to a shortening of the O−O bond in coordinated $H_2O_2$. Specifically, the O−O distances (1.422(5) and 1.443(5) Å in **1**, and 1.445(4) Å in **2**) are shorter than those in crystalline $H_2O_2$ (1.461(3) Å)[26] and cesium hexahydroperoxo stannate (1.482(2) Å)[14]. In addition, the two O−O fragments of the $H_2O_2$ ligands in **1** are almost

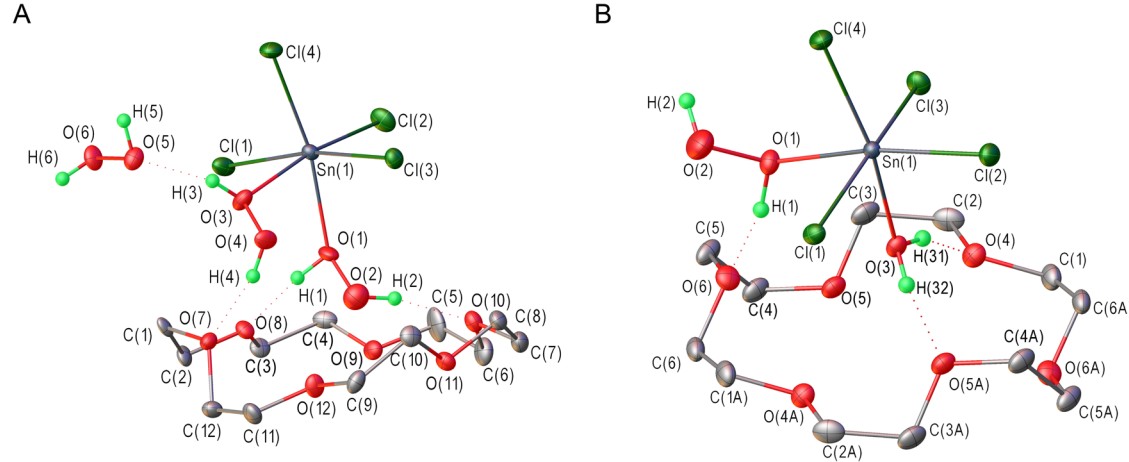

**Fig. 2 | The crystal structures of hydrogen peroxide complexes with tin tetrachloride. A** symmetric unit in **1**. **B** Asymmetric unit in **2**. 18-crown-6 molecule lies on crystallographic inversion center. Displacement ellipsoids are shown at a 50% probability level. Hydrogen bonds are represented by dotted lines. The H atoms of the macrocyclic ether are omitted for clarity.

**Table 1 | Selected geometric parameters obtained by scXRD analysis and DFT calculation (gas phase), and QTAIM-derived energetics for contacts involving the $H_2O_2$ ligand in 1 and 2 at the ωB97X-D3/TZVPP level of theory[a]**

| Cpd | d(Sn–O₂H₂), Å | | $E_{Sn-O}$,[b] kcal mol⁻¹ | Contact | d(D…A), Å | | d(H…A), Å | | $E_{int}$,[b] kcal mol⁻¹ | $ΣE_{int}$, kcal mol⁻¹ |
|---|---|---|---|---|---|---|---|---|---|---|
| | X-ray | DFT | | | X-ray | DFT | X-ray | DFT | | |
| **1** | 2.179(4) | 2.261 | 16.4 | O(1)H⋯O(8) | 2.583(5) | 2.642 | 1.74(3) | 1.683 | 9.7 | 17.8 |
| | | | | O(2)H⋯O(10) | 2.671(5) | 2.733 | 1.82(3) | 1.761 | 8.1 | |
| | 2.200(3) | 2.323 | 13.4 | O(3)H⋯O(5) | 2.542(5) | 2.690 | 1.74(3) | 1.726 | 8.9 | 16.4 |
| | | | | O(4)H⋯O(7)[c] | 2.730(5) | 3.110 | 1.91(3) | 2.419 | 2.0[d] | |
| | | | | O(4)H⋯O(12) | 3.156(5) | 2.837 | 2.60(4) | 1.945 | 5.5 | |
| **2** | 2.225(3) | 2.358 | 12.3 | O(1)H⋯O(6) | 2.548(4) | 2.631 | 1.70(3) | 1.704 | 9.4 | 13.4 |
| | | | | O(2)H⋯Cl(3)[d] | 3.114(4) | 3.142 | 2.33(4) | 2.267 | 4.0 | |

[a]See Fig. 2 for atom labeling.
[b]Calculated using Eq. 1 (see "Methods" section of the main text).
[c]O(7) and O(12) were neighboring oxygen atoms, and peroxide O(4)H switched its position between these O atoms in the DFT calculations.
[d]The intramolecular O(2)–H⋯Cl(4) hydrogen bond was identified in the calculated structure.

parallel to each other (Supplementary Fig. 5). However, remarkably, there is no hydrogen bond between the $H_2O_2$ ligands, despite these neighboring molecules bearing acidic protons. Moreover, the interligand O⋯O distances in **1** (O(1)⋯O(3) = 2.844(5) Å and O(2)⋯O(4) = 3.019(6) Å) are shorter than the sum of their van der Waals radii (3.04 Å), which could indicate the presence of a weak contact (e.g., a chalcogen bond).

## DFT calculations

To study the mutual influence of various types of bonding of $H_2O_2$ ligands we performed gas-phase DFT calculations for **1** and **2**. As mentioned above, the coordination of $H_2O_2$ with Sn$^{IV}$ increases the acidity of $H_2O_2$, which increases the strength of the hydrogen bond that subsequently forms. According to DFT calculations, the sum of the energies of two hydrogen bonds formed by $H_2O_2$ molecules acting as proton donors correlates with the Sn−O distance, i.e., the high sum corresponds to the short distance, revealing the synergy between the coordination of an $H_2O_2$ ligand and its hydrogen bonding with the second coordination sphere (Table 1).

To highlight the synergy of primary (Sn−O) and secondary (hydrogen bond) interactions in an $H_2O_2$−Sn complex, we studied the interaction of an $SnCl_4(H_2O_2)_2$ complex with imidazole, which served as a model of proton-donating and proton-accepting molecules in the second coordination sphere. The imidazole moiety of histidine (His42)

plays an important role in the peroxidase catalytic cycle[27] contributing to the formation of the supposed iron-hydrogen peroxide complex with heme to give [Fe-OOH] form - a so-called Compound 0 - at the next step[27,28]. DFT calculations were performed for $SnCl_4(H_2O_2)_2$, $H_2O_2 \cdot C_3H_4N_2$, $SnCl_4(H_2O_2)_2 \cdot C_3H_4N_2$, and analogs containing the imidazolium cation ($C_3H_5N_2^+$), e.g., $SnCl_4(H_2O_2)_2 \cdot C_3H_5N_2^+$ (Table 2, Supplementary Fig. 6). The optimized structure of the $SnCl_4(H_2O_2)_2 \cdot C_3H_4N_2$ adduct features the O–H⋯N hydrogen bond with an N⋯O distance of 2.622 Å, which correlates well with the corresponding interaction in the reported crystal structure of histidine peroxosolvate[29]. Adduct $SnCl_4(H_2O_2)_2 \cdot C_3H_5N_2^+$ contains an $H_2O_2$ ligand functioning as an acceptor of the acidic proton of the imidazolium cation, with an N⋯O distance of 2.967 Å.

In the $SnCl_4(H_2O_2)_2 \cdot C_3H_4N_2$ adduct, the hydrogen bonding of coordinated $H_2O_2$ with imidazole leads to the shortening of the Sn−O distance for the hydrogen-bonded $H_2O_2$ ligand. The N⋯O distance of the hydrogen bond of the $H_2O_2$ ligand is also shorter than that in imidazole peroxosolvate, $H_2O_2 \cdot C_3H_4N_2$. In contrast, the hydrogen bonding of peroxide oxygen with imidazolium in $SnCl_4(H_2O_2)_2 \cdot C_3H_5N_2^+$ causes a substantial elongation of the corresponding Sn−O bond. This reflects the mutual influence of primary (Sn−O) and secondary (hydrogen) bonds that is further supported by the energy analysis. The value and sign of the cooperative effect, $ΔΔH_{coop}$, for the $SnCl_4(H_2O_2)_2$ adduct with a second-coordination-sphere proton acceptor (imidazole molecule) or

**Table 2 | Bond distances (d, in Å) and energies (in kcal mol⁻¹) for the hydrogen and coordination bonds in $SnCl_4(H_2O_2)_2$, $H_2O_2 \cdot C_3H_4N_2$, $SnCl_4(H_2O_2)_2 \cdot C_3H_4N_2$, $H_2O_2 \cdot C_3H_5N_2^+$, and $SnCl_4(H_2O_2)_2 \cdot C_3H_5N_2^+$ at the ωB97X-D3/TZVPP level of theory**

| | $SnCl_4(H_2O_2)_2$ | | $H_2O_2 \cdot C_3H_4N_2$ | | $SnCl_4(H_2O_2)_2 \cdot C_3H_4N_2$ | | |
|---|---|---|---|---|---|---|---|
| $\Delta H_f$ [a] | −8.99 (−4.12*) | | −6.36 | | −19.88 (−15.01*) | | |
| Contact | d | $E_{int}$ [b] | d | $E_{int}$ | d | $E_{int}$ | $\Delta\Delta H_{coop}$ [c] |
| Sn–O1 | 2.371 | 11.7 | – | – | 2.284 | 15.1 | −3.06 |
| Sn–O3 | 2.334 | 13.1 | – | – | 2.339 | 12.9 | |
| N···(H)O2 | – | – | 1.742 | 8.3 | 1.608 | 10.9 | |

| | | $H_2O_2 \cdot C_3H_5N_2^+$ | | $SnCl_4(H_2O_2)_2 \cdot C_3H_5N_2^+$ | | |
|---|---|---|---|---|---|---|
| $\Delta H_f$ | | −2.54 | | −11.52 (−6.66*) | | |
| Contact | | d | $E_{int}$ [b] | d | $E_{int}$ | $\Delta\Delta H_{coop}$ |
| Sn–O1 | | – | – | 2.432 | 9.9 | 1.06 |
| Sn–O3 | | – | – | 2.337 | 13.0 | |
| N(H)···O2 | | 1.837 | 7.0 | 1.988 | 4.9 | |

* Energy relative to $SnCl_4(H_2O_2)_2$.

[a] Formation enthalpies, $\Delta H_f$, calculated relative to isolated reactants.

[b] Bond energies, $E_{int}$, calculated using Eq. 1 (see "Methods" section of the main text).

[c] $\Delta\Delta H_{coop}$, calculated using Eq. 2 (see "Methods" section of the main text).

**Table 3 | Distances between each $Sn^{IV}$ coordination center and its ligand's hydrogen-bond acceptor**

| Compound | Method | d(Sn···A) (A = O, Cl, or N), Å | | |
|---|---|---|---|---|
| | | $H_2O_2$ | | $H_2O$ |
| | | Proximal | Distal | |
| $[SnCl_4(H_2O_2)_2]\cdot H_2O_2\cdot 18$-crown-6 (**1**) | scXRD | 4.217 (O) 4.274 (O) | 4.867 (O) 4.548 (O) | – |
| $2[SnCl_4(H_2O_2)(H_2O)]\cdot 18$-crown-6 (**2**) | scXRD | 4.281 (O) | 5.315 (Cl) | 4.031 (O) 4.339 (O) |
| $2[SnCl_4(H_2O)_2]\cdot 18$-crown-6 (**3**) | scXRD | – | – | 4.011 (O) 4.415 (O) 4.187 (O) 4.664 (Cl) |
| $SnCl_4(H_2O_2)_2\cdot C_3H_4N_2$ | DFT | – | 4.695 (N) | – |

proton donor (imidazolium cation) estimated by Eq. 2 was used as a measure of the synergism or antagonism of primary and secondary interactions[30].

A preliminary attempt to protonate $SnCl_4(H_2O_2)_2$ in gas-phase DFT calculations led to the dissociation of this complex. Therefore, we expected that the interaction of a proton donor with an $H_2O_2$ ligand would lead to a decrease in the stability of the complex. Indeed, hydrogen bonding of imidazolium to a distal oxygen has an antagonistic energetic effect ($\Delta\Delta H_{coop} = 1.06$ kcal mol$^{-1}$). In contrast, a synergistic effect was found between $H_2O_2$ coordination and its proton donation to the imidazole fragment ($\Delta\Delta H_{coop} = -3.06$ kcal mol$^{-1}$).

The distances between the $Sn^{IV}$ coordination center and the ligand hydrogen-bond acceptor in the scXRD data of **1–3** and the calculated adduct $SnCl_4(H_2O_2)_2\cdot C_3H_4N_2$ are presented in Table 3. As expected, the distances between $Sn^{IV}$ and the hydrogen-bond acceptor of the aqua ligand correlate with those for the proximal hydroxo group of the $H_2O_2$ ligand. However, the distance between the coordination center and the acceptor of the distal hydroxo group of the $H_2O_2$ ligand is always longer than that for the aqua ligand when the acceptor is of the same type. This observation also calls for a speculation on the coordination of $H_2O_2$ to enzymes' heme, which occurs in aqueous systems. The $Fe\cdots N^{His42}$ distance in peroxidases is approximately 5.7 Å[31,32], which is too long for activation of $H_2O$ but is suitable for activation of $H_2O_2$. In this hydrophobic pocket, the $OH\cdots N$ hydrogen bonding of the distal OH group to His42 should stabilize the binding of $H_2O_2$ to a heme Fe. One can speculate that due to this unique hydrogen bond of the distal proton, the enzyme can differentiate between $Fe\cdot OH_2$ and $Fe\cdot O_2H_2$ complexes, as the cooperative effect (as estimated herein; Table 2) would overcome a stronger Fe–O bond with $H_2O$ than with $H_2O_2$ and stabilize the encounter $[Fe-O_2H_2]$ complex. Furthermore, as the coordination to an Fe ion increases the acidity of the proximal OH group, it should facilitate deprotonation of Compound 0 yielding $[Fe-OOH]$ hydroperoxo complex.

The scarcity of structurally resolved $H_2O_2$ complexes has hindered an examination of the structure and bonding of $H_2O_2$ ligands in reaction intermediates and life-sustaining biocomplexes. This limitation is attributable to $H_2O_2$ being less basic than the common coordinating solvents by which it is replaced in a metal coordination sphere. In addition, $H_2O_2$ is poorly soluble in most non-coordinating solvents. In this study, we suggested a synthetic approach based on the use of pure $H_2O_2$ as both solvent and ligand. Thus, coordinatively unsaturated $SnCl_4$ effectively binds $H_2O_2$ yielding $SnCl_4(H_2O_2)_2$, which was characterized by $^{119}Sn$ and $^{17}O$ NMR spectroscopy. This complex comprises rather strong Sn-O bonds (12–16 kcal mol$^{-1}$ according to DFT analysis), but its $H_2O_2$ ligands could be easily substituted in the $Sn^{IV}$ coordination sphere by molecules of higher basicity, i.e., MeOH, MeCN, and even $H_2O$. The use of 18-crown-6 as a bulky yet stable $H_2O_2$-proton acceptor stabilized $SnCl_4(H_2O_2)_2$ as its 18-crown-6 adduct (**1**). The addition of $H_2O$ gave the stepwise substitution products **2** and **3**.

ScXRD analysis of these complexes revealed their rich set of non-covalent interactions, including the shortest $O(H)\cdots O$ distances in hydrogen bonds formed by $H_2O_2$ in known crystal structures. Complemented by the results of our DFT analysis, this demonstrates the synergistic effects of a coordination bond with $Sn^{IV}$ and hydrogen bonding with a second coordination sphere on the properties of an $H_2O_2$ ligand. The energies of two hydrogen bonds formed by each $H_2O_2$ ligand acting as a proton donor correlate with the Sn–O distance in **1** and **2**, with the higher hydrogen-bond energy value corresponding to the shorter Sn–O distance. Remarkably, none of the $H_2O_2$ ligands participated in hydrogen bonding as proton acceptors, despite the proximity of acidic protons. Our DFT study of model $SnCl_4(H_2O_2)_2$ adducts with imidazole/imidazolium suggested that this is due to the antagonistic energetic effect of such interactions.

In summary, this study demonstrated that second-coordination-sphere hydrogen bonding plays a key role in the stabilization of $H_2O_2$ coordination. The non-covalent interactions of $H_2O_2$ ligands not only contribute to the total energy of the system but also increase the basicity of the $H_2O_2$ ligand, which enhances coordination bonding. This explains why $H_2O_2$ coordination, despite being impossible in aqueous solution under equilibrium conditions, is common in nature, such as in oxygenases. Coordination with a Lewis acid has previously been proposed to be a key factor in the activation of $H_2O_2$ for the oxidation of organic substrates[5]. Therefore, we envisage prospects for the development of new catalytic systems in which the distance between the coordination center and the hydrogen bond acceptor is approximately 5 Å. This would make it possible to utilize the synergism of the primary and secondary interactions and ensure the coordination of $H_2O_2$ in the presence of $H_2O$ or other polar molecules.

## Methods
### Synthesis of anhydrous $H_2O_2$ and $SnCl_4$
Caution! Working with concentrated $H_2O_2$ and chlorine is hazardous and requires appropriate precautions to be taken.

Small amounts of anhydrous $H_2O_2$ can be obtained from its crystalline adducts with organic compounds[33,34]. However, this requires the use of organic solvents (diethyl ether or MeCN) that may absorb $H_2O$ and other impurities, and also may remain in the product $H_2O_2$ and thus interact with $SnCl_4$ in the next step. Therefore, in the current study, we purified commercial $H_2O_2$ via a two-stage vacuum distillation process. First, 30 wt.% $H_2O_2$ was distilled under vacuum to remove stabilizers and other impurities and afford 18 wt.% pure aqueous $H_2O_2$. Second, this $H_2O_2$ solution was concentrated by rectification under vacuum, controlling the boiling by passing argon (Ar), to afford 99.9 wt.% $H_2O_2$ (as determined by permanganometry; Supplementary Methods).

As commercial $SnCl_4$ may contain impurities that can catalyze $H_2O_2$ decomposition, we synthesized $SnCl_4$ from ultrapure metal Sn by chlorination followed by rectification (Supplementary Methods).

### NMR spectroscopy
The solutions for NMR experiments were prepared in an Ar-filled glovebox ($O_2$ and $H_2O$ concentrations < 0.1 ppm) and then immediately placed in the spectrometer (Supplementary Methods). The time between the preparation of the solutions and the NMR experiments did not exceed 10 min. $^{17}O$ and $^{119}Sn$ NMR spectra ($\delta$, ppm) were collected at 303 K on a Bruker AVANCE III 600 spectrometer operating at 81.36 MHz and 223.79 MHz, respectively. $^{17}O$ and $^{119}Sn$ chemical shifts were referenced to $H_2O$ and tetramethyltin, respectively. NMR spectra were processed using TopSpin software.

### ScXRD
Single crystals of **1–4** that were suitable for X-ray analysis were collected from the corresponding mother liquors without additional recrystallization, placed on microscope slides, and then coated with a

perfluorinated oil (Fomblin YR-1800). Subsequently, appropriate single crystals were mounted on MicroMeshes™ (MiTeGen) and then immediately positioned beneath a cold stream of nitrogen on the diffractometer, which was a Bruker D8 Venture instrument that used graphite monochromatized molybdenum K-alpha radiation ($\lambda = 0.71073$ Å) and was operated in $\omega$-scan mode at 100 K. Absorption corrections based on measurements of equivalent reflections were applied[35]. The structures were solved by direct methods and refined by full matrix least-squares on $F^2$ with anisotropic thermal parameters for all non-hydrogen atoms[36]. The hydrogen atoms of $H_2O_2$ and $H_2O$ molecules in 1–3 were found from difference Fourier synthesis and refined with distance restraints. The hydrogen atoms of 18-crown-6 in 1–3 and MeCN solvent in 4 were placed in idealized positions and refined using a riding model. Additional crystallographic data for 1–4 are provided in the Supplementary Information (Supplementary Figs. 1–4, Supplementary Tables 1–5).

ScXRD of 1 indicated a relatively short distance between H(4) and O(12), i.e., 2.60(4) Å. However, it was much longer than a typical O–H···O hydrogen bond, and its O–H···O angle (124.7°) lay outside the normal range. Thus, this contact was not attributable to a hydrogen bond and appeared to be a forced contact due to crystal packing.

The X-ray structure of 4 exhibited better resolution (see Supplementary Table 1) than that of a previously reported structure[17].

## DFT calculations

Clusters containing the fragments of asymmetric units of 1 and 2 were taken from the corresponding scXRD data, and calculations were performed using various approaches (Supplementary Methods, Supplementary Data 1). Optimization at the $\omega$B97X-D3/TZVPP level of theory afforded the best correlation between the calculated Sn–O distances and those obtained from scXRD (Table 1) and is therefore used in the discussion of the results below. The quantum theory of atoms in molecules was applied to analyze the electron density parameters at the O–H···O hydrogen-bond critical points (Table 1 and Supplementary Table 6).

In these calculations, $H_2O$ was used as a solvent (in the conductor-like polarizable continuum model approach) because it closely approximates the solvent properties of $H_2O_2$ (i.e., its dielectric constant and acidity/basicity) and because polar solvents typically weaken non-covalent interactions, meaning that the detection of a pronounced effect in such a solvent serves as strong evidence of the proposed concept.

The energies of non-covalent interactions in the optimized clusters of 1 and 2 were estimated according to Espinosa's approach[37] (Eq. 1) and are presented in Table 1.

$$E_{int}[\text{kcal mol}^{-1}] = 269.2 G_b[\text{atomic units}] \quad (1)$$

where $G_b$ is a Lagrangian of kinetic energy density at the bond critical point.

The cooperative effect, $\Delta\Delta H_{coop}$, was calculated according to ref. 30:

$$\Delta\Delta H_{coop}(ABC) = H_{ABC} - (H_{AB} + H_{BC} + H_{AC}) + (H_A + H_B + H_C) \quad (2)$$

where $H$ are enthalpies of the corresponding trimer ($SnCl_4(H_2O_2)_2\cdot C_3H_{4+n}N_2^{n+}$), dimers ($SnCl_4(H_2O_2)_2$, $H_2O_2\cdot C_3H_{4+n}N_2^{n+}$ and $SnCl_4(H_2O_2)//C_3H_{4+n}N_2^{n+}$) and monomers ($SnCl_4(H_2O_2)$, $H_2O_2$ and $C_3H_{4+n}N_2^{n+}$) at the trimer geometry; $n = 0$ for imidazole, and $n = 1$ for imidazolium.

## Data availability

The X-Ray crystallographic data for the structures reported in this Article have been deposited at the Cambridge Crystallographic Data Centre (CCDC) under deposition numbers CCDC 2260843 (1), 2260844 (2), 2260845 (3) and 2260846 (4). These data can be obtained free of charge via https://www.ccdc.cam.ac.uk/structures/. The equilibrium Cartesian coordinates data generated in this study are provided as the Supplementary Data 1. All data are available in the main text, the Supplementary Information and from the corresponding authors.

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

## Acknowledgements

This work was supported by the Russian Science Foundation (grant no. 22-13-00426, https://rscf.ru/en/project/22-13-00426/). A.A.K. and M.V.B. acknowledge the support from the City University of Hong Kong (Project 7006013). O.L. acknowledges the Israel Science Foundation (grant number 1215/19) for financial support. The X-ray diffraction studies were performed using the equipment of the JRC PMR IGIC RAS. The NMR studies were performed using the equipment of the CKP FMI IPCE RAS. A.G.M. and P.V.P. acknowledge Dr. A.V. Churakov for helpful discussion.

## Author contributions

P.V.P. conceived the project; the project was supervised by P.V.P., O.L. and M.V.B.; P.V.P., M.V.B. and O.L. gathered the fundings and contributed equally. Conceptualization of the synthesis and NMR spectra assignment was provided by P.V.P.; A.A.M. and P.A.E. prepared the hydrogen peroxide; A.G.M. and P.A.E. prepared the solutions for NMR studies and performed synthesis of the crystals; A.G.M. conducted scXRD experiment and described the crystal structures; E.S.B., G.A.K and Yu.G.G. performed the NMR studies; O.A.F., N.V.B. and E.S.S. conducted DFT calculations and provided discussion of the DFT results; M.N.B. performed $SnCl_4$ synthesis and purification; A.A.K. analyzed the literature on the hydrogen peroxide complexes; All authors discussed the results; P.V.P., M.V.B., and O.L. wrote the paper with input from all of the authors.

## Competing interests

The authors declare no competing interests.
