## [Peer Review File · Nature Communications]

Synergism of primary and secondary interactions in a crystalline hydrogen peroxide complex with tinREVIEWER COMMENTS

Reviewer #1 (Remarks to the Author):

This article is focused on the reactivity of pure hydrogen peroxide with SnCl₄ and characterization of resulting complexes, which include two crystal structures of complexes bearing molecular hydrogen peroxide ligand, which is extremely rare. The characterization of these complexes with ¹¹⁹Sn and ¹⁷O NMR and scXRD is nice, but I was surprised that no other characterization was provided for these compounds. Typically, some form of bulk characterization to supplement single-crystal measurements is required for novel compounds.

The article claims that hydrogen peroxide complexes are interesting due to Compound 0, which is reasonable. However, going so far as to describe these tin(IV) chloride complexes dissolved in pure hydrogen peroxide as mimics for Compound 0 is a pretty big stretch. The use of DFT to further explore these complexes is interesting and adds some connection to Compound 0, but I cannot comment on the appropriateness of the specific theoretical methods to study these complexes.

The impact of the paper heavily relies on the claim of the first crystal structures of metal complexes bearing hydrogen peroxide with localized hydrogen atoms, which I do not think is a reasonable claim. The previously published structure of a zinc hydrogen peroxide complex must have localized hydrogen atoms. The author claims that the position of the protons in that structure could not be determined precisely due to the mixed occupancy issue, but the implication that one or more protons could have moved to the secondary sphere of that complex is unreasonable because it would require protonation of a sulfonyl oxygen atom, which is unprecedented and not supported by pK_a data. If there are additional doubts as to the structural details of the previously published zinc hydrogen peroxide compound, then a crystallographer with experience in second-sphere hydrogen bonding should be included among the reviewers, as I am not an expert in that area.

In summary, I find this work impressive and interesting, but I question whether the impact is high enough for this journal and I find the characterization data lacking.

Additional comments

- In Fig 2A the upper left structure contains an unexpected O-O bond between the methanol ligands that is not mentioned in the narrative. I assume this is simply a mistake.
- Line 102-103, the authors state that hydrogen peroxide “does not form homogeneous solutions with nonpolar solvents,” which is somewhat misleading. Hydrogen peroxide is known to dissolve in diethyl ether.

- Line 128. The authors state that bound hydrogen peroxide will not appear in ^{17}O NMR due to broadening and assigns free H_2O_2 to 185.3 ppm. However, in line 136 a signal at 183.9 ppm is assigned to bound H_2O_2 in a direct contradiction.
- In line 133, the bridging aqua complex should not be described as a dimer.
- There are multiple complexes presented in Fig 2A that have no characterization. They seem to be reasonable suggestions, but the only evidence of their existence are small shifts in the NMR spectra. This does not seem sufficient to confirm the presence and specific structure of those complexes, so they should be more clearly represented as possible (but unconfirmed) complexes. Furthermore, some form of bulk characterization of compounds 1-4 should be provided in addition to single-crystal data. Is it possible to collect IR data on any of these complexes?
- The use of crown ethers to provide hydrogen bonding for bound H_2O_2 is a great approach and the crystal structures are impressive.

Reviewer #2 (Remarks to the Author):

The review is attached.

Although crystallographic characterization of the hydrogen peroxide–tin tetrachloride 18-crown-6 complexes is undoubtedly important (given the well-known weak ability of H₂O₂-ligand to coordinate), I do not think the presented results warrant publication in the *Nature Commun.* Instead, I would recommend reporting these data in more specialized and expert-oriented journals such as *Inorganic Chemistry* or *Organometallics*.

Some concrete issues also contribute to the above-mentioned general opinion:

1) The composition of the paper is unbalanced: very long “Introduction” (3 pages) and exceptionally long (1.5 pages) “Conclusions” vs. rather short “Results and Discussion” part (8 pages, of which 3 pages are computational discussion).

2) The “Introduction” of the paper is rather confusing. An unnecessarily comprehensive description of the biochemical aspects of H₂O₂-coordination to Fe, including discussion on Compound 0, has quite nothing to do with the experimental results presented in this paper (apart from the very last paragraph of DFT calculations). Why such an irrelevant discussion is given in the Introduction, where the readers expect to be provided with the setting of the scene? It looks like the “Introduction” and the “Results and Discussion” are from different papers.

3) Figure 2A: numbering of all compounds presented and discussed in this scheme is strictly required.

4) Figure 2A: it seems there should be no bond between two oxygen atoms in the methanol adduct.

5) Caption of Figure 3: symmetry operations, which are related to the centrosymmetric space groups and thus do not give valuable information, should be deleted.

6) SI, Experimental part, page S3, line 66: what is “sulfochromic acid”? Is it a chromic acid reagent, that is a mixture of sulfuric acid and dichromate? Please specify.

Reviewer #3 (Remarks to the Author):

In the current work, the authors presented a new approach to obtain stable H₂O₂-metal complexes using SnCl₄ and H₂O₂ as both solvent and ligand. Using DFT calculations, the authors arrived at the conclusion that the second coordination sphere of a catalytic base (proton acceptor) is key to stabilizing the H₂O₂-metal complex through cooperative effect. The results are novel and significant, and I recommend the publication of the manuscript after addressing the following comments:

1. In Eq 2, there seems to be a term from the subtraction, i.e., the $\Delta H_f(\text{C}_3\text{H}_4 + n\text{N}_2 + \text{SnCl}_4(\text{H}_2\text{O}_2))$. It is the formation enthalpy between the imidazole/imidazolium and SnCl₄(H₂O₂), and accounts for the interaction between the imidazole/imidazolium with a SnCl₄ coordinated by a single H₂O₂. In my opinion, this term should be taken into account when calculating cooperativity among the three interacting species: H₂O₂, SnCl₄(H₂O₂), and C₃H₄+nN₂, because any two of the three interacting species have a pairwise interaction energy. This term is currently missing from the subtraction in Eq 2. If added, it will likely change the ΔH_{coop} results in Table 2, which is the major evidence from DFT calculations supporting the authors' claim that the second coordination sphere stabilizes the H₂O₂-metal complex through cooperative effect.

2. The authors need to cite and briefly discuss the following paper in their discussion: J. Phys. Chem. B 2023, 127, 51, 10987–10999. This paper discusses the binding stability of H₂O₂ at the active site of a heme-thiolate enzyme, and the conclusion of the paper is consistent with the current manuscript in that the hydrogen bond environment around the H₂O₂ stabilizes the binding.

REPLIES TO REVIEWERS' COMMENTS

Reviewer #1 (Remarks to the Author):

Recommendation: Publish after minor revisions.

Comments:

>>This article is focused on the reactivity of pure hydrogen peroxide with SnCl₄ and characterization of resulting complexes, which include two crystal structures of complexes bearing molecular hydrogen peroxide ligand, which is extremely rare. The characterization of these complexes with ¹¹⁹Sn and ¹⁷O NMR and scXRD is nice, but I was surprised that no other characterization was provided for these compounds. Typically, some form of bulk characterization to supplement single-crystal measurements is required for novel compounds.

We thank Reviewer #1 for wise and constructive remarks and comments. Please find our point-by-point replies below.

Answer: Neat hydrogen peroxide is relatively dense solvent (1.45 g/ml) with high boiling point (150 °C). It is impossible to quickly separate crystals of **1** and **2** from the mother liquor containing a high concentration of hydrogen peroxide because hydrogen peroxide does not evaporate readily (due to the low vapor pressure) or mix with non-coordinating solvents inert to compounds **1** and **2**, and the use of coordinating solvents leads to the destruction of these complexes. Although the mother liquor residues should stabilize the crystals, their presence complicates spectral studies and elemental analysis. Additionally, the hydrogen peroxide decomposes slowly when in contact with perfluorinated oil and equipment parts. Water formed during the decomposition of hydrogen peroxide decomposes the complexes **1** and **2**. Despite our best efforts, we were unable to obtain reproducible X-ray powder diffraction patterns of **1** and **2**. The combination of these factors makes scXRD the only relevant research method. The single crystal X-ray diffraction experiments were reproduced several times from different syntheses (the additional cif file for compound **1** attached with revised article as Related Manuscript File). In our opinion, the consistency between the scXRD and ¹¹⁹Sn and ¹⁷O NMR data in solution indicates that crystals **1** and **2** were not random reaction products.

Additionally, we performed the bulk characterization of compounds **3** and **4** and added the results to the ESI: powder X-Ray diffractogram (Supplementary Fig. 5), elemental analysis (tin, carbon and hydrogen content), FTIR spectroscopy (Supplementary Fig. 6, Supplementary Table 7) for compound **3** and powder X-Ray diffractogram for compound **4** (Supplementary Fig. 7).

>>The article claims that hydrogen peroxide complexes are interesting due to Compound **0**, which is reasonable. However, going so far as to describe these tin(IV) chloride complexes dissolved in pure hydrogen peroxide as mimics for Compound **0** is a pretty big stretch. The use of DFT to further explore these complexes is interesting and adds some connection to Compound **0**, but I cannot comment on the appropriateness of the specific theoretical methods to study these complexes.

Answer: The point about the analogy with Compound **0** was further emphasized by the Editor. To address it we have changed the title of the manuscript to: “**A crystalline hydrogen peroxide complex with tin: synergism of primary and secondary interactions**” and have revised the Introduction and Conclusion to remove the description of the similarities between tin complexes

and Compound 0 (including Fig. 1). Actually, we primarily emphasized not the similarity of the tin complex with Cpd0, but the similarity of the interaction pattern in Cpd0 and complex 1.

>>The impact of the paper heavily relies on the claim of the first crystal structures of metal complexes bearing hydrogen peroxide with localized hydrogen atoms, which I do not think is a reasonable claim. The previously published structure of a zinc hydrogen peroxide complex must have localized hydrogen atoms. The author claims that the position of the protons in that structure could not be determined precisely due to the mixed occupancy issue, but the implication that one or more protons could have moved to the secondary sphere of that complex is unreasonable because it would require protonation of a sulfonyl oxygen atom, which is unprecedented and not supported by pKa data. If there are additional doubts as to the structural details of the previously published zinc hydrogen peroxide compound, then a crystallographer with experience in second-sphere hydrogen bonding should be included among the reviewers, as I am not an expert in that area.

Answer: Due to the disordering of the O atoms and tosyl fragments in the crystal structure of the hydrogen peroxide complex with zinc [Wallen, C. M., Bacsa, J. & Scarborough, C. C. Hydrogen Peroxide Complex of Zinc. *J. Am. Chem. Soc.* **137**, 14606–14609 (2015).], the protons cannot be localized objectively from the Difference Fourier synthesis and, therefore, were placed in geometrically idealized positions and refined using the riding model. However, we completely agree with the Reviewer that undoubtedly the previously published Zn-H₂O₂ complex is the first structurally characterized H₂O₂ coordination compound and proton transfer from molecular hydrogen peroxide to sulfonamide ligands is not possible. In our case, the objectively localized protons of coordinated hydrogen peroxide in the structures of **1** and **2** allow to undoubtedly discriminate some of the short O...O distances as belonging to hydrogen bonds. In accordance with the Reviewer's comments, the sentence "structures **1** and **2** are the first crystal structures of hydrogen peroxide complexes with objectively localized protons" both from the abstract and the text of the manuscript. We have also changed the phrase "However, the isomorphous substitution of H₂O₂ with H₂O (in a 50:50 occupancy ratio) disordered the O atoms and tosyl fragments in this complex, preventing the positions of the H₂O₂ protons being determined unequivocally." to "However, the isomorphous substitution of H₂O₂ with H₂O (in a 50:50 occupancy ratio) disordered the O atoms and tosyl fragments in this complex, preventing the positions of the H₂O₂ protons being determined objectively". We also underscored, in the revised article, the pioneering way by which hydrogen peroxide is used in this research as a solvent and a ligand. This is a general new way to overcome solvent competitive ligation with H₂O₂.

>>In summary, I find this work impressive and interesting, but I question whether the impact is high enough for this journal and I find the characterization data lacking.

Answer: We thank the reviewer again for positive evaluation of our work, and we believe that all of the reviewers' suggestions were incorporated into the revised article.

>>Additional comments

•In Fig 2A the upper left structure contains an unexpected O-O bond between the methanol ligands that is not mentioned in the narrative. I assume this is simply a mistake.

Answer: Figure 1 (former Fig 2) has been modified and the O-O bond between methanol ligands was eliminated.

•Line 102-103, the authors state that hydrogen peroxide “does not form homogeneous solutions with nonpolar solvents,” which is somewhat misleading. Hydrogen peroxide is known to dissolve in diethyl ether.

Answer: We agree with the reviewer's comment. Diethyl ether is generally considered a nonpolar solvent due to its low dipole moment (1.15D, [C. Reichardt, T. Welton *Solvents and Solvent Effects in Organic Chemistry* 4th Edn. (2010) pp. 549–586. <https://doi.org/10.1002/9783527632220.app1>]). Anyway, diethyl ether is a coordinating solvent and interacts with tin tetrachloride forming $\text{SnCl}_4(\text{OEt}_2)_2$, as it was previously proven by ^{119}Sn NMR studies [Fărcașiu, D., Leu, R. & Ream, P. J. The 1:1 and 2:1 complexes of diethyl ether with tin tetrachloride and their stability, studied by ^{119}Sn NMR spectroscopy. *J. Chem. Soc. Perkin Trans. 2* 427–432 (2001). doi:10.1039/b004849f]. Additionally, the crystal structure of complex $\text{SnCl}_4(\text{OEt}_2)_2$ was previously characterized by scXRD [Yatsenko, A. V., Aslanov, L. A., Burtsev, M. Y. & Kravchenko, E. A. The crystal and molecular structure and ^{35}Cl NQR spectra of $\text{SnCl}_4(\text{Et}_2\text{O})_2$. *Russ. J. Inorg. Chem* **36**, 1147–1149 (1991)]. We have tried to dissolve SnCl_4 in diethyl ether but the addition of SnCl_4 to diethyl ether resulted in immediate precipitation of $\text{SnCl}_4(\text{Et}_2\text{O})_2$ complex which was confirmed by powder XRD (Figure I). Experimental diffractogram of the isolated precipitate corresponds quite well with theoretical one simulated from the previously reported scXRD for $\text{SnCl}_4(\text{OEt}_2)_2$ [Yatsenko, A. V., Aslanov, L. A., Burtsev, M. Y. & Kravchenko, E. A. The crystal and molecular structure and ^{35}Cl NQR spectra of $\text{SnCl}_4(\text{Et}_2\text{O})_2$. *Russ. J. Inorg. Chem* **36**, 1147–1149 (1991)]. The small low-angle shift of the experimental reflections relative to the simulated diffraction pattern was attributed to displacement of the wet sample due to shrinkage during measurement. Thus, it seems that diethyl ether cannot be used to obtain hydrogen peroxide complexes. To clarify our statement, we have substituted the phrase “does not form homogeneous solutions with nonpolar solvents” by “does not form homogeneous solutions with non-coordinating solvents” in the manuscript.

Figure I. Experimental (a) and theoretical (b) powder XRD diffractograms of $\text{SnCl}_4(\text{Et}_2\text{O})_2$ complex.

•Line 128. The authors state that bound hydrogen peroxide will not appear in ^{17}O NMR due to broadening and assigns free H_2O_2 to 185.3 ppm. However, in line 136 a signal at 183.9 ppm is assigned to bound H_2O_2 in a direct contradiction.

Answer: We thank the reviewer for this comment. This regrettable mistake appeared during the text editing process, and we are very grateful to the reviewer for correcting it. Indeed, the signal

at 183.9 ppm. in the spectrum ^{17}O (line 136) corresponds to free hydrogen peroxide. As is mentioned in the manuscript (lines 128-130), “an H_2O_2 ligand in a complex contains two non-equivalent neighboring O atoms and thus its ^{17}O NMR signals are quadrupole broadened and therefore not detectable.” We corrected the corresponding phrase and now it reads:

” Addition of H_2O to give a 0.5 H_2O -to- SnCl_4 molar ratio resulted in $[\text{SnCl}_4(\text{H}_2\text{O}_2)]_2(\mu\text{-H}_2\text{O})$ being the dominant species (Fig. 1Bd), and its ^{17}O NMR spectrum contained slightly upfield-shifted signals for free H_2O_2 ($\delta_{\text{O}} = 183.9$ ppm) and a bridging aqua ligand signal ($\delta_{\text{O}} = 51.3$ ppm; Fig. 1Cd).”

•In line 133, the bridging aqua complex should not be described as a dimer.

Answer: We thank the reviewer for this comment. We removed the term ‘dimer’ from the text.

•There are multiple complexes presented in Fig 2A that have no characterization. They seem to be reasonable suggestions, but the only evidence of their existence are small shifts in the NMR spectra. This does not seem sufficient to confirm the presence and specific structure of those complexes, so they should be more clearly represented as possible (but unconfirmed) complexes.

Answer: The ^{119}Sn NMR in high field spectrometer is very sensitive to the Sn coordination environment and can be used for characterization of Sn(IV) complexes in solution even when the ligands are chemically similar like H_2O_2 , $\mu\text{-H}_2\text{O}$ or H_2O . The ^{119}Sn NMR spectra presented in Figure 1Bg demonstrate that at least four different tin(IV) complexes are present in the same solution, and the high-field signal corresponds to the diaqua complex which ^{119}Sn NMR spectrum has been previously reported. Additionally, the spectra at Figures 1B b and c contain two separate signals corresponding to two different tin(IV) forms. These ^{119}Sn NMR spectra reveal five different tin(IV) complexes. Furthermore ^{17}O NMR spectra confirm the bridging and terminal H_2O coordination with tin(IV) depending on the ratio between H_2O and tin tetrachloride. Therefore we believe that there are enough experimental evidences for the assignments presented in the text and Figure 1A. The scXRD data support the NMR assignments, since three of these five complexes detected by NMR present in crystal structures **1–3** isolated from the corresponding solutions.

•Furthermore, some form of bulk characterization of compounds 1-4 should be provided in addition to single-crystal data. Is it possible to collect IR data on any of these complexes?

Answer: Unfortunately, despite our best efforts, we were unable to provide bulk characterization of the compounds **1** and **2** due to stability issues (please see our answer on first comment). The characterization of compounds **3** and **4** were added to ESI.

•The use of crown ethers to provide hydrogen bonding for bound H_2O_2 is a great approach and the crystal structures are impressive.

Answer: We thank reviewer for his constructive comments and valuable remarks which helped us improve the article.

Reviewer #2 (Remarks to the Author):

Although crystallographic characterization of the hydrogen peroxide–tin tetrachloride 18-crown-6 complexes is undoubtedly important (given the well-known weak ability of H₂O₂-ligand to coordinate), I do not think the presented results warrant publication in the Nature Commun. Instead, I would recommend reporting these data in more specialized and expert-oriented journals such as *Inorganic Chemistry* or *Organometallics*.

Some concrete issues also contribute to the above-mentioned general opinion:

Answer: We thank Reviewer #2 for his/her constructive criticisms and comments. Please find our point-by-point replies below.

1) The composition of the paper is unbalanced: very long “Introduction” (3 pages) and exceptionally long (1.5 pages) “Conclusions” vs. rather short “Results and Discussion” part (8 pages, of which 3 pages are computational discussion).

Answer: following reviewer’s comment we have significantly shortened the “Introduction” (from 979 to 615 words) and balanced “Conclusions” and “Results and Discussion” sections.

2) The “Introduction” of the paper is rather confusing. An unnecessarily comprehensive description of the biochemical aspects of H₂O₂-coordination to Fe, including discussion on Compound 0, has quite nothing to do with the experimental results presented in this paper (apart from the very last paragraph of DFT calculations).

Why such an irrelevant discussion is given in the Introduction, where the readers expect to be provided with the setting of the scene? It looks like the “Introduction” and the “Results and Discussion” are from different papers.

Answer: We thank reviewer for pointing this out. Following the reviewer’s comment, we completely changed the Introduction section to match the main findings. We have shortened the discussion regarding Compound 0 in the “Discussion” and removed it completely from the “Concluding remarks” and Abstract.

3) Figure 2A: numbering of all compounds presented and discussed in this scheme is strictly required.

Answer: We would like to ensure the reviewer that all the complexes from the Figure 2A are unambiguously described in the text. Complexes that were characterized by X-ray diffraction analysis were numbered Crystal 1-4. All other complexes were mentioned in the description as SnCl₄(H₂O₂)₂, SnCl₄(H₂O₂)₂(μ-H₂O), SnCl₄(H₂O₂)(H₂O), [SnCl₄(H₂O)]₂(μ-H₂O), SnCl₄(MeOH)₂ and SnCl₄(MeCN)₂.

4) Figure 2A: it seems there should be no bond between two oxygen atoms in the methanol adduct.

Answer: We thank the reviewer for this comment. This regrettable mistake appeared during the editing, and we are very grateful to the reviewer for correcting it. The Figure 1A (former Figure 2A) has been modified and the accidental O-O bond between methanol ligands has been erased.

5) Caption of Figure 3: symmetry operations, which are related to the centrosymmetric space groups and thus do not give valuable information, should be deleted.

Answer: We thank the reviewer for noticing. The Figure 2 (former Figure 3) caption was corrected. For the moment, Fig. 2. shows the asymmetric units of **1** (A) and **2** (B). The sentence, "18-crown-6 ether molecule lies on crystallographic inversion centre" has been added to the figure caption. The symmetry operations were deleted from figure caption.

6) SI, Experimental part, page S3, line 66: what is "sulfochromic acid"? Is it a chromic acid reagent, that is a mixture of sulfuric acid and dichromate? Please specify.

Answer: Yes, the sulfochromic acid is a mixture of concentrated sulfuric acid and dichromate. We removed this term from the text and changed it to the "dichromate" since it catalyzes the hydrogen peroxide decomposition.

We thank the reviewer for the careful reading and thoughtful comments.

Reviewer #3 (Remarks to the Author):

>>In the current work, the authors presented a new approach to obtain stable H₂O₂-metal complexes using SnCl₄ and H₂O₂ as both solvent and ligand. Using DFT calculations, the authors arrived at the conclusion that the second coordination sphere of a catalytic base (proton acceptor) is key to stabilizing the H₂O₂-metal complex through cooperative effect. The results are novel and significant, and I recommend the publication of the manuscript after addressing the following comments.

Answer: We thank the reviewer for recommending the manuscript for publication, recognizing correctly the generality of the dual use of hydrogen peroxide as a solvent and ligand and for very helpful and constructive comments.

>> In Eq 2, there seems to be a term from the subtraction, i.e., the $dH_f(C_3H_{4+n}N_{2n^+} \& SnCl_4(H_2O_2))$. It is the formation enthalpy between the imidazole/imidazolium and SnCl₄(H₂O₂), and accounts for the interaction between the imidazole/imidazolium with a SnCl₄ coordinated by a single H₂O₂. In my opinion, this term should be taken into account when calculating cooperativity among the three interacting species: H₂O₂, SnCl₄(H₂O₂), and C₃H_{4+n}N_{2n⁺}, because any two of the three interacting species have a pairwise interaction energy. This term is currently missing from the subtraction in Eq 2. If added, it will likely change the ΔH_{coop} results in Table 2, which is the major evidence from DFT calculations supporting the authors' claim that the second coordination sphere stabilizes the H₂O₂-metal complex through cooperative effect.

Answer: We appreciate the reviewer sharp-eye for identifying the calculation error.

Actually, we initially apply the simplified formula, without long range AC interaction, but which takes into account interactions in the *separated* dimers, which typically works for simple hydrogen bonded systems.

Indeed, the correct formula for estimating cooperativity in the 3-body system will be:

$$\Delta H_{coop}(ABC) = H_{ABC} - (H_{AB} + H_{BC} + H_{AC}) + (H_A + H_B + H_C)$$

where H are enthalpies of corresponding trimer (SnCl₄(H₂O₂)₂•C₃H_{4+n}N_{2ⁿ⁺}), dimers (SnCl₄(H₂O₂)₂, H₂O₂•C₃H_{4+n}N_{2ⁿ⁺} and SnCl₄(H₂O₂)/C₃H_{4+n}N_{2ⁿ⁺}) and monomers (SnCl₄(H₂O₂),

H₂O₂ and C₃H_{4+n}N₂ⁿ⁺) at the trimer geometry. (Note that we apply the monoperoxide tin complex as a monomer A in these calculations).

With this formula we get the somewhat smaller, but still pronounced, cooperativity effect ($\Delta\Delta H_{\text{coop}} = -3.06$ kcal/mol) for imidazole complex and *anticooperative* (antagonistic) effect for imidazolium one ($\Delta\Delta H_{\text{coop}} = +1.06$ kcal/mol), which gives even better support to our hypothesis.

The presence of the antagonistic effect is also supported by the results of scXRD, as none of the hydrogen peroxide ligands in the structures of **1** and **2** acts as a proton acceptor.

>> The authors need to cite and briefly discuss the following paper in their discussion: J. Phys. Chem. B 2023, 127, 51, 10987–10999. This paper discusses the binding stability of H₂O₂ at the active site of a heme-thiolate enzyme, and the conclusion of the paper is consistent with the current manuscript in that the hydrogen bond environment around the H₂O₂ stabilizes the binding.

Answer: We thank the reviewer for this comment. We have added another article (J. Phys. Chem. B. 2023, 127, 8809–8824; 10.1021/acs.jpcc.3c04589 – reference #7 in the revised manuscript) of the same author and published in the same journal which the reviewer probably had in mind.

We thank the reviewer for recommending publication after the revision and for wise and constructive remarks. We are very grateful for the opportunity to correct an error that affects the calculation part.

REVIEWERS' COMMENTS

Reviewer #1 (Remarks to the Author):

I am largely satisfied with the edits made in response to reviewer comments.

The only suggestion I have left is to more clearly distinguish between characterized crystal compounds 1-4 and the other reasonably assigned, but not fully characterized compounds in Figure 1A. The structural assignments might be supported by ^{119}Sn -NMR, but that is still only one form of characterization. Perhaps the figure caption could be modified to something like the following:

...Coordination of tin tetrachloride (SnCl_4) with hydrogen peroxide (H_2O_2), formation of crystalline compounds 1–4, and assumed intermediate solvato compounds supported by ^{119}Sn -NMR.

Also, I think it would be helpful to number these compounds, which I think is what Reviewer 2 suggests in their comment 3 (just with the incorrect figure number), for ease of reference in the text and supporting information.

Reviewer #3 (Remarks to the Author):

The authors have addressed my previous comments.

REPLIES TO REVIEWERS' COMMENTS

Reviewer #1 (Remarks to the Author):

Recommendation: Publish after minor revisions.

Comments:

>> I am largely satisfied with the edits made in response to reviewer comments.

The only suggestion I have left is to more clearly distinguish between characterized crystal compounds 1-4 and the other reasonably assigned, but not fully characterized compounds in Figure 1A. The structural assignments might be supported by ^{119}Sn -NMR, but that is still only one form of characterization. Perhaps the figure caption could be modified to something like the following:

...Coordination of tin tetrachloride (SnCl_4) with hydrogen peroxide (H_2O_2), formation of crystalline compounds 1–4, and assumed intermediate solvato compounds supported by ^{119}Sn -NMR.

Also, I think it would be helpful to number these compounds, which I think is what Reviewer 2 suggests in their comment 3 (just with the incorrect figure number), for ease of reference in the text and supporting information.

We thank Reviewer #1 for wise and constructive remarks and comments.

Answer: We agree with the reviewer's comment and have changed the figure caption taking also into account the editor's note. New caption is "**Fig. 1.** Complexation of tin tetrachloride in hydrogen peroxide solution supported by NMR spectroscopy. (A) Coordination of tin tetrachloride (SnCl_4) with hydrogen peroxide (H_2O_2), formation of crystalline compounds **1-4** and assumed intermediate complexes supported by ^{119}Sn NMR." We have numbered SnCl_4 complexes according to the reviewer's comments.

We thank the reviewer again for positive evaluation of our work, and we believe that all of the reviewers' suggestions were incorporated into the revised article.

Reviewer #3 (Remarks to the Author):

>> The authors have addressed my previous comments.

Answer: We thank the reviewer for their helpful remarks and for positive perception of our work.